# Time-Aware Prior Fitted Networks for Zero-Shot Forecasting with Exogenous Variables

## Abstract

In many forecasting settings, the target series comes with exogenous covariates: promotions and prices for retail demand, temperature for energy load, calendar/holiday flags for traffic or sales, and grid load or fuel costs for electricity prices. Ignoring such exogenous covariates can seriously degrade forecasting accuracy, especially when they signal phase changes or spikes in the target series. Most current time-series foundation models (e.g., `Chronos`, `Sundial`, `TimesFM`, `TimeMoE`, `TimeLLM`, and `LagLlama`) ignore exogenous covariates and make forecasts solely from the time-series history, limiting their performance. In this paper we focus on bridging this gap by developing `ApolloPFN`, a prior-data fitted network (PFN) that is time-aware (unlike prior PFNs) and that natively incorporates exogenous covariates (unlike prior univariate forecasters). Our design introduces two major advances: (i) a synthetic data generation procedure tailored to resolve the failure modes that arise when tabular (non-temporal) PFNs are applied to time-series, and (ii) time-aware architectural modifications that embed the inductive biases needed to fully exploit the time-series context. We demonstrate that `ApolloPFN` achieves state-of-the-art results across benchmarks containing *exogenous* information such as M5 and electric price forecasting.

## 1 Introduction

In many high-impact forecasting scenarios, leveraging *exogenous* information, i.e. inputs beyond the raw target time-series values, is essential. For example, in electricity price forecasting and consumer demand forecasting, information about planned prices and promotions, merchandising changes, holidays and local events, weather forecasts, and competitor pricing, are naturally encoded categorically and can shift demand sharply. Ignoring this information often induces large, systematic errors as seen in Figure 1. In spite of the value of exogenous information, the vast majority of current time-series foundation models (TSFMs) such as `Chronos` (Ansari et al., 2024), `Sundial` (Liu et al., 2025), `TimesFM` (Das et al., 2023), `TimeMoE` (Shi et al., 2025), `TimeLLM` (Jin et al., 2024), and `LagLlama` (Rasul et al., 2023) cannot handle exogenous covariates directly, or they require fine-tuning on the data (Arango et al., 2025; Wang et al., 2024; Potapczynski et al., 2024a). Fine-tuning is often undesirable as it adds runtime, complicates the inference pipeline, increases deployment costs, and weakens the anonymity and isolation of downstream customer data. Therefore, a practical TSFM should be able to natively incorporate accompanying exogenous covariates when they are available.

There are a few foundation-like models that accept exogenous covariates in a zero-shot setting: in particular, `TabPFN-TS` (Hoo et al., 2025) and `Moirai` (Woo et al., 2024). Assessing `Moirai`'s true zero-shot capability is complicated as it was exposed to almost all public time-series benchmarks (including large-scale suites such as `Gift-Eval` Aksu et al. (2024)) during training; therefore, finding benchmarks with non-overlapping training and testing observations is difficult. Even so, it often ranks below `TabPFN-TS`, even against the benchmarks it was trained on. Crucially, though, `TabPFN-TS` is *not* a time-series model *per se*—instead, it simply appends a handful of time-series features to a tabular foundation model. Therefore, it lacks core temporal inductive biases. As we discovered and describe below, the central problem is that the architecture of `TabPFN-TS` is invariant to the order of the data. Order invariance is a reasonable inductive bias in the tabular i.i.d. case, but it is *not* a reasonable inductive bias for the time-series context, where the arrow of time defines an important ordering. In practice, this bias leads to characteristic failure modes when

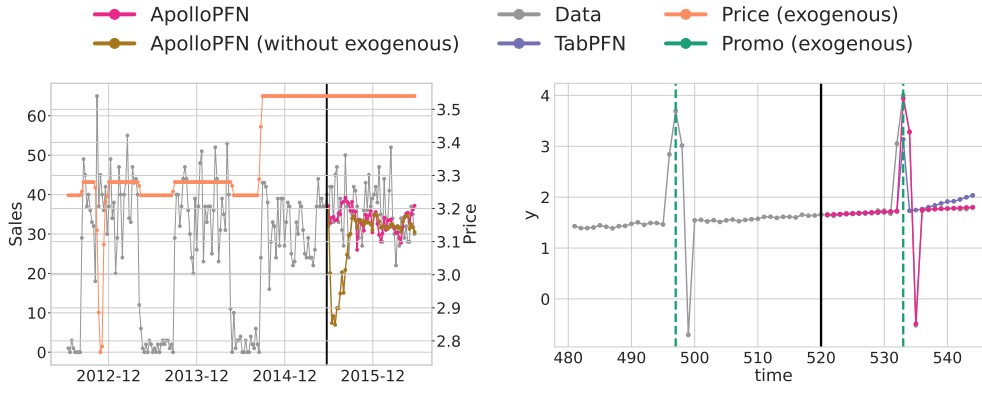

(a) M5 weekly sales

(b) time-series with promotional activity

Figure 1: (a) **Not using exogenous information leads to catastrophic forecasting errors.** We compare the predictions of `ApolloPFN` with and without using exogenous information for the weekly sales of a real product from the M5 benchmark. Ignoring the rise in price leads the forecaster to predict a decreased demand as previously observed in the past (brown). In contrast, using exogenous information, the same model predicts a higher demand (red). (b) **Prior-data fitted networks such as `TabPFN-TS` fail to capture ordered patterns.** We compare the prediction of `TabPFN-TS` and `ApolloPFN` for a synthetic time-series that has a recurrent pattern of a ramp-up period before a promotion, a spike on the promotion, a ramp-down period, and then a subsequent decrease in demand. The exogenous promotion information is encoded as a binary indicator. Train data is to the left of the black line and forecasts are to the right.

forecasting with `TabPFN-TS`. Such failure modes include disregarding order-dependent patterns, inability to work well across unseen frequencies, weak trend extrapolation, insufficient emphasis on recent context, and poorly calibrated confidence intervals.

In this paper, we show how to effectively leverage exogenous variables for zero-shot forecasting, including the following contributions:

- We provide a detailed characterization of the shortcomings of existing PFNs such as `TabPFN-TS` *for time-series forecasting*. In particular, we show that `TabPFN-TS` has intrinsic limitations due to the i.i.d. assumption that informs how the synthetic training data is generated, as well as its architectural specification. For these reasons, it fails to understand temporal autocorrelations, making it challenging to accurately predict ordered patterns, as illustrated in Figure 1 (b). Based on these findings, we argue that existing PFNs are *not* suitable as *time-series* FMs (Section 3.)

- We introduce `ApolloPFN`, a model that circumvents the limitations of `TabPFN-TS` through a novel data generation procedure and architectural choices (Section 4). This consists of two complementary components. First, we introduce a synthetic data generation procedure for time-series that relies on a new graph generation algorithm (which accelerates learning as seen in Figure 3) coupled with time-dependent root nodes (see Section 4.1). Second, we incorporate inductive biases into our architecture that reflect the importance of order in time-series data (Section 4.2) and present several ablations on real and synthetic data to bolster our choices (Section 4.3).

- We extensively compare `ApolloPFN` against SOTA baselines, including `TabPFN-TS` and `Moirai`, in several datasets spanning more than 90K time series that have accompanying exogenous covariates, demonstrating the broad effectiveness of `ApolloPFN`. (Section 5.)

## 2 BACKGROUND

### 2.1 NOTATION

Since our method is based on a tabular foundation model `TabPFN` (Hollmann et al., 2023; 2025), our notation refers tabular datasets in some contexts, and to time series in others. In the tabular context the data is indexed by $i$ as $\mathcal{D}_{\text{train}} = (\boldsymbol{x}_i, y_i)_{i=1}^{N_{\text{train}}}$ where we would make predictions for $(y_i)_{i=1}^{N_{\text{test}}}$ using $\mathcal{D}_{\text{train}}$ and the covariates $(\boldsymbol{x}_i)_{i=1}^{N_{\text{test}}}$. In contrast, when forecasting, we index our data by $t$ as $\mathcal{D}_{\text{train}} = (\boldsymbol{x}_t, y_t)_{t=1}^{T}$ where we therefore have $T$ previous time steps as history and would make predictions

for a horizon $H$ $(y_t)_{t=T+1}^{T+H}$ using all of $\mathcal{D}_{\text{train}}$ and the future covariate information $(\boldsymbol{x}_t)_{t=T+1}^{T+H}$ (when available). Most of the neural forecasters in the literature solely provide predictions of the form $(y_T, \ldots, y_{T+H}) = f_\theta(y_1, \ldots, y_T)$, ignoring all $\boldsymbol{x}_t$. However, as seen in Figure 1 (a), the covariates provide crucial information to maintain accurate predictions. In this paper, we will provide a model that makes predictions of the type $(y_T, \ldots, y_{T+H}) = f_\theta(y_1, \ldots, y_T, \boldsymbol{x}_1, \ldots, \boldsymbol{x}_{T+H})$ for varying $T$ and $F$, where $F$ is the covariate dimensionality $\boldsymbol{x}_t \in \mathbb{R}^F$.

## 2.2 PFNs

Müller et al. (2022; 2025) introduced a novel paradigm to perform Bayesian inference through prior-fitted networks (PFNs). First, a user defines an algorithm to sample datasets $\mathcal{D}_{\text{train}} = (\boldsymbol{x}_i, y_i)_{i=1}^{N_{\text{train}}}$ usually by sampling a vector or graph $\xi \sim p(\xi)$ and then sampling $(\boldsymbol{x}_i, y_i) \sim p(\boldsymbol{x}, y|\xi)$. By defining a neural network $q_\theta$ that minimizes the following loss

$$\mathcal{L}(\theta) = - \mathbb{E}_{p(\boldsymbol{x}, y)} \log q_\theta(y_{\text{test}}|\boldsymbol{x}_{\text{test}}, \mathcal{D}_{\text{train}})$$

the neural network $q_\theta(y_{\text{test}}|\boldsymbol{x}_{\text{test}}, \mathcal{D}_{\text{train}})$ approximates the posterior predictive distribution (PPD) $p(y_{\text{test}}|\boldsymbol{x}_{\text{test}}, \mathcal{D}_{\text{train}})$ directly (Müller et al., 2022). The key insight is that by having the neural network $q_\theta$ approximate the PPD, we circumvent the need to approximate a high-dimensional posterior $p(\xi|\mathcal{D}_{\text{train}})$ or define a closed-form likelihood $p(y|\boldsymbol{x}, \xi)$, which is how the PPD is usually computed: $p(y_{\text{test}}|\boldsymbol{x}_{\text{test}}, \mathcal{D}_{\text{train}}) = \int p(y_{\text{test}}|\boldsymbol{x}_{\text{test}}, \xi) p(\xi|\mathcal{D}_{\text{train}}) d\xi$ (Murphy, 2012; Hoffman & Gelman, 2014; Wilson & Izmailov, 2020).

The data creation in `TabPFN` (Hollmann et al., 2023; 2025) is illustrative of how a user can generate implicit priors through sampling. `TabPFN` uses structured causal models (SCMs) which are directed acyclical graphs (DAG) where the nodes $z_i$ are defined by the relationship with their parent nodes $\text{PA}(i)$ as $z_i = f_i(z_{\text{PA}(i)}) + \epsilon_i$ where $f_i$ is some function and $\epsilon_i$ is measurement noise. To generate SCMs, at a high level, Hollmann et al. (2025) samples DAGs from the random growing networks with preferential attachment process from Krapivsky & Redner (2023) and then defines $f_i$ as either MLPs (with distinct activations), categorical functions or decision trees (with distinct depths). To generate $N$ observations, we pass random noise to the root nodes and propagate the values through the graph in topological order. We then pick some $F$ nodes and set them as the features $\boldsymbol{x}_i \in \mathbb{R}^F$ and a node as $y_i \in \mathbb{R}$ for each $i = 1, \ldots, N$. See Section 4.1 and Appendix B for more details on the synthetic data generation.

The architecture in `TabPFN` (Hollmann et al., 2025) closely resembles the transformer architecture from Radford et al. (2019). Given a tensor $\boldsymbol{Z} \in \mathbb{R}^{N \times F \times D}$ where $N$ is the number of observations (both train and test, $N = N_{\text{train}} + N_{\text{test}}$), $F$ the number of features and $D$ the embedding dimension, we have that the main blocks of the `TabPFN` architecture work as follows

$$\boldsymbol{Z} \leftarrow \text{LN}_1^{(\ell)}(\boldsymbol{Z} + \text{AttnFeat}^{(\ell)}(\boldsymbol{Z}))$$
$$\boldsymbol{Z} \leftarrow \text{LN}_2^{(\ell)}(\boldsymbol{Z} + \text{AttnSamp}^{(\ell)}(\boldsymbol{Z})) \quad (1)$$
$$\boldsymbol{Z} \leftarrow \text{LN}_3^{(\ell)}(\boldsymbol{Z} + \text{MLP}^{(\ell)}(\boldsymbol{Z}))$$

for $\ell = 1, \ldots, L$ layers. Appendix A explains how we embed the input data $(y_i)_{i=1}^{N_{\text{train}}}$ and $(\boldsymbol{x}_i)_{i=1}^{N}$ into $\boldsymbol{Z}$. The first and second operations are variants of the classical attention mechanism (Vaswani et al., 2017), $\text{LN}(\cdot)$ stands for layer normalization (Ba et al., 2016) and $\text{MLP}(\cdot)$ is a MLP applied to the embedding dimension. `AttnFeat` assumes the $F$ axis is the variable part of the mechanism, the $D$ axis is the embedding, and the remaining axes are batch axes. In contrast, `AttnSamp` assumes that the $N$ axis is the variable part of the mechanism, the $D$ axis is the embedding and the remaining axes are also treated as batch axes. Moreover, the attention matrix $\boldsymbol{A}_{f,:,:} \in \mathbb{R}^{N \times N}$ is going to avoid interactions between test points that we are trying to fill-in. That is, $\boldsymbol{A}_{f,i,j} = 0$ if both $i$ and $j$ belong to test indices.

The previous architecture thus allows for a variable number of observations $N$ and a variable number of features $F$. Moreover, as no positional encodings are used for `AttnSamp` the mechanism is permutation invariant, which is sensible for i.i.d. data.

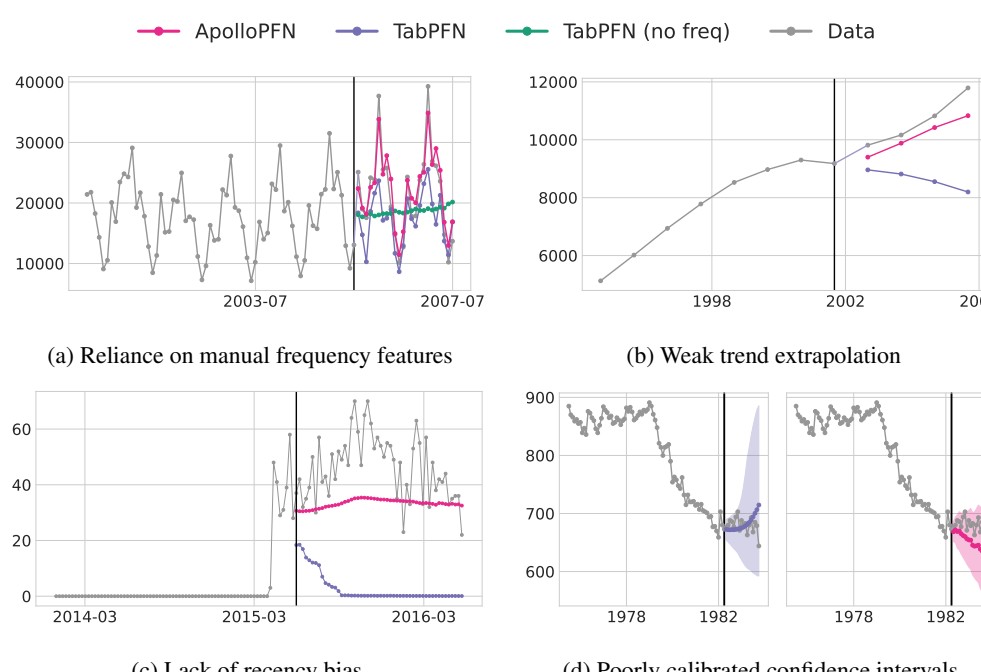

(a) Reliance on manual frequency features

(b) Weak trend extrapolation

(c) Lack of recency bias

(d) Poorly calibrated confidence intervals

Figure 2: **Failure modes of `TabPFN-TS` for time-series data that `ApolloPFN` addresses.** We show some illustrative examples of each failure case with different real time-series: we use a time-series in Tourism Monthly for (a), in Tourism Yearly for (b), in M5 Weekly for (c) and in M1 Monthly for (d). In the plots, the train data is to the left of the black line and the forecasts to the right. *(a)* When `TabPFN-TS` is not given frequency features it predicts an average of prior history (green line). In contrast, `TabPFN-TS` might capture some time patterns when frequency features are available but miss others outside the frequency range (it does not capture the largest spikes). *(b)* `TabPFN-TS` has problems extrapolating trends especially in short context cases. *(c)* The predictions of `TabPFN-TS` erroneously revert back to zero as that is the most common value in the context. *(d)* The range of the 90% confidence intervals in `TabPFN-TS` substantially increases to capture previously seen values rather than reflect the uncertainty over the trend of the time series.

## 3 FAILURE MODES OF TABPFN-TS

`TabPFN-TS` (Hoo et al., 2025) introduces a series of manually engineered time-series features into the tabular foundational model `TabPFN-v2` (Hollmann et al., 2025) in order to make forecasts. Although `TabPFN-TS` achieves competitive performance on several time-series forecasting benchmarks, it exhibits fundamental failure modes due to the absence of time-series specific inductive biases, raising concerns about the deployment of such models in industry-critical applications.

**Inability to learn ordered patterns.** Ordered seasonal patterns that span across multiple time steps are very common in industry applications such as demand forecasting (where a product has a gradual increase in demand until its promotion date and sharply drops after it) and energy consumption (where usage steadily builds up toward peak hours and then declines overnight). These types of patterns are not purely cyclical, but instead they reflect structured temporal dependencies that unfold over multiple horizons. An example of such a pattern is shown in Figure 1(b), which shows that `TabPFN-TS` cannot capture in-context a sequence of events as it lacks temporal inductive biases. Instead, the model resolves to outputting a smaller spike in the promotional event.

**Dependency on manually engineered frequency features.** `TabPFN-TS` relies on a running index feature as well as frequency features that are taken from the timestamp of the data (such as day-of-week, day-of-month, month-of-year, etc.) or estimated frequencies obtained through a FFT decomposition of the time-series (Hoo et al., 2025). That is, $x_{t,j} = \sin(2\pi \frac{\tau(t)}{P_j})$ or $x_{t,j} = \cos(2\pi \frac{\tau(t)}{P_j})$ where for example, in the case of day-of-week $\tau(t) \in \{1, \cdots, 7\}$ and $P_j = 7$ and so forth. As seen in Figure 2(a), if the frequencies are not used then `TabPFN-TS` only estimates the mean of the previous observations. However, `TabPFN-TS` makes accurate predictions when the relevant

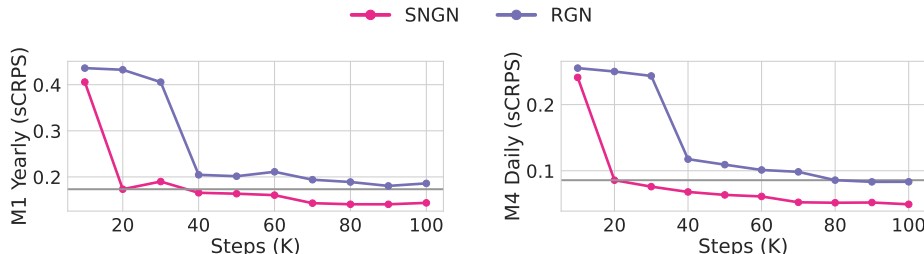

Figure 3: **Our graph generation algorithm accelerates learning**. We compare the test benchmark performance of our ApolloPFN model trained with the random growing network algorithm (RGN) and our single node growing network algorithm (SNGN) at different training steps. With SNGN we achieve better performance at 20K iterations than at 80K with RGN.

frequencies are explicitly included in the data, but it struggles to capture patterns that do not align with regular calendar structures.

**Weak trend extrapolation.** Already noted in (Hoo et al., 2025), TabPFN-TS demonstrates a limited ability to extrapolate time-series trends as seen in Figure 2(b). This phenomenon most likely results from the model's inability to consider the order of the data when estimating the trend.

**Lack of a recency bias.** TabPFN-TS treats all historical time points equally when making predictions. Many applications operate in environments with constant distribution shifts, e.g., the underlying data changes over time due to factors like promotions, policy changes, or macroeconomic conditions. Accurately predicting under these distribution shifts is critical for a reliable deployment of time-series models. Figure 2(c) shows that TabPFN-TS struggles to capture a sudden uptick in demand, failing to forecast based on the most recent observations.

**Poorly calibrated confidence intervals.** TabPFN-TS produces confidence intervals that emphasize the entire historical context rather than weighting observations according to their consistency with the prevailing trend in the time series. Figure 2(d) clearly shows this phenomenon where the huge confidence interval simply reflects values obtained in the distant past. This failure undermines trust and complicates decision-making in industry critical time-series applications.

The underlying reason as to why TabPFN-TS suffers from the aforementioned failure modes is because it was trained and developed for i.i.d. data! While the model incorporates some time-series-specific features, it fails to capture relevant temporal relationships in the data.

## 4 APOLLOPFN

We now present the architectural and data interventions that allows us to develop ApolloPFN, a PFN model that leverages the order and temporal relationship of time-series data.

### 4.1 TEMPORAL TABLES

To follow the TabPFN training procedure in Hollmann et al. (2025), we have to create synthetic tabular data in the following manner. First, we sample a DAG $\mathcal{G} \sim p(\mathcal{G})$ via random growing networks (RGN) with preferential attachment (Krapivsky & Redner, 2023) (see Algorithm 1). The graph $\mathcal{G}$ then determines the parent nodes PA($j$) for each node $j$ in $\mathcal{G}$. We then define the following structural causal model (SCM) as in Pearl (2009): $V_j = f_j(V_{\text{PA}(j)}) + \epsilon_j$ where $f_j$ is either a MLP, a categorical encoding or a decision tree, and $\epsilon_j$ is measurement noise. In this context, the different nodes $j$ in $\mathcal{G}$ represent different features with their relationships given by the SCM. The graphs $\mathcal{G}$ that we sample using Algorithm 1 are characterized by having several root nodes and short paths as seen in Figure 6 (top) in Appendix B.1.

Then, to generate a tabular dataset $\mathcal{D}_{\mathcal{G}} = (\boldsymbol{x}_i, y_i)_{i=1}^N$ with $\boldsymbol{x}_i \in \mathbb{R}^F$ and $y_i \in \mathbb{R}$, we first sample the numbers of observations that we need $N \sim p(N)$ as well as the features $F \sim p(F)$. Once $N$ and $F$ are determined, we then start sampling i.i.d. noise $v_{i,r} \sim p(\eta)$ for each root note $r$ in $\mathcal{G}$ and for each $i = 1, \ldots, N$, and then propagate these root values in topological order through the SCM such that $v_{i,j} = f_j(v_{i,j_1}, \ldots, v_{i,j_k}) + \epsilon_i$ where PA($j$) = $\{j_1, \ldots, j_k\}$ for all $j$ in $\mathcal{G}$. Once we obtain

$(v_{i,1}, \ldots, v_{i,|\mathcal{G}|})_{i=1}^{N}$, where $|\mathcal{G}|$ denotes the numbers of nodes in $\mathcal{G}$, we then randomly select $F + 1$ features (excluding root nodes) and set $x_{i,j} = v_{i,\pi(j)}$ and $y_i = v_{i,\pi(F+1)}$ where $\pi(\cdot)$ represents the random selection.

There are two key modifications that we introduce to the previous synthetic data generation procedure. First, we develop a new graph generation algorithm (Algorithm 2), named single node growing network (SNGN), which generates graphs with a single node and various paths that connect the nodes, as seen in Figure 6 (bottom). More importantly, as seen in Figure 3, our use of SNGN dramatically increases the speed at which the model starts to make accurate predictions. For Figure 3, we trained two different `ApolloPFN` models, one with RGN and one with SNGN, leaving the rest of the hyperparameters fixed. We then evaluated the performance of the model checkpoints every 10K iterations on different benchmarks. We consistently see the model trained with SNGN achieves a better performance faster than the model trained with RGN. See Appendix B for details.

Then, we sample the values of root nodes $(v_{t,r})_{t=1}^{T}$ through some stochastic process, thereby introducing a time dependency. In particular, we make the root nodes a combination between a sine and cosine function with randomly sampled frequencies $(\phi_1^{(r)}, \phi_2^{(r)})$ and amplitudes $(\alpha_1^{(r)}, \alpha_2^{(r)})$. That is, $v_{t,r} = \alpha_1^{(r)} \sin(\phi_1^{(r)} t) + \alpha_2^{(r)} \cos(\phi_2^{(r)} t)$ for all $t = 1, \ldots, T$. As a result, we now generate datasets $\mathcal{D}_{\mathcal{G}} = (\boldsymbol{x}_t, y_t)_{t=1}^{T}$ where nearby values like $y_{t+1}$ are correlated with $y_t$, and so on, in contrast to sampling root nodes as $v_{i,r}$ independently for each $i$. After we define the temporal root nodes, we then propagate the values in the graph to obtain the rest of the features, as in Hollmann et al. (2025). We still follow the input normalization procedure from `TabPFN`. That is, we z-score the data $(y_t)_{t=1}^{T}$ before passing it to the model and then we invert the z-scoring when outputting the predictions $(y_t)_{t=T+1}^{T+H}$. Note that our mean $\mu_T$ and standard deviation $\sigma_T$ only depend on the data up to $T$ to avoid leaking future information.

## 4.2 ARCHITECTURAL MODIFICATIONS

### 4.2.1 POSITIONAL ENCODINGS

Once we have a data generation procedure that has a time dependency, it then makes sense to introduce an inductive bias to the attention mechanism that reflects these time relationships. A natural choice is to incorporate RoPE embeddings (Su et al., 2023) to the attention mechanism in `AttnSampl`$^{(\ell)}(\cdot)$ because RoPE would then make the keys and query interactions obey $\boldsymbol{q}_{t+h}^{\mathsf{T}} \boldsymbol{R}_h \boldsymbol{k}_t$, where $\boldsymbol{R}_h$ is a weight matrix such that $\boldsymbol{q}_{t+h}^{\mathsf{T}} \boldsymbol{R}_h \boldsymbol{k}_t \to 0$ as $h \to \infty$. In other words, the keys and queries of nearby observations are weighted more highly.

RoPE solely incorporates a notion of relative distance between the observations. To incorporate an absolute notion we use a similar construction to Vaswani et al. (2017) and define absolute positional encodings of the form $\boldsymbol{\Omega} \in \mathbb{R}^{T \times D}$

$$\Omega_{t,2d+1} = \sin\left(2\pi t \frac{2^{2d+1}}{2^{12}}\right) \quad \text{and} \quad \Omega_{t,2d} = \cos\left(2\pi t \frac{2^{2d}}{2^{12}}\right)$$

which we add to $\boldsymbol{Z}_f \leftarrow \boldsymbol{Z}_f + \boldsymbol{\Omega}$ for all $f = 1, \ldots, F$ (see Equation 1).

### 4.2.2 EXPANDING ATTENTION

Given that `TabPFN` (Hollmann et al., 2023; 2025) was trained on i.i.d. data, a key modification in the attention mechanism of AttnSampl$^{(\ell)}(\cdot)$ is that test observations do not attend to each other but only to the train observations. Therefore when making $M$ predictions for $(\boldsymbol{x}_j)_{j=1}^{N_{\text{test}}}$ we use the PPD of the form $p(y_j | \boldsymbol{x}_j, (\boldsymbol{x}_i, y_i)_{i=1}^{N})$ for each $j = 1, \ldots, N_{\text{test}}$ independently of each other. However, in the case of time-series, if we are to make $H$ predictions we require that all future *exogenous* information (if present) then informs the current predictions. In other words, we expect that $p(y_{T+h} | (\boldsymbol{x}_t)_{t=T+1}^{T+H}, (\boldsymbol{x}_t, y_t)_{t=1}^{T})$ for all $h = 1, \ldots, H$. To achieve the previous relationship we simply allow all points to attend to each other on AttnSampl$^{(\ell)}(\cdot)$.

### 4.3 IMPACT OF MODIFICATIONS

In Figure 2 we observe how our new time-series synthetic data generation process coupled with the architectural changes presented in the previous sections enables `ApolloPFN` to resolve the failure modes of `TabPFN-TS`. In Figure 4 we perform an ablation to show the performance improvement when training `ApolloPFN` with only our time-dependent data `ApolloPFN(-)`, then training the model with positional encodings `ApolloPFN(RoPE)` and, finally, allowing the attention mechanism to learn interactions between all the predictions `ApolloPFN(RoPE+Full)`. The baseline for Figure 4 is `TabPFN-TS` (Hollmann et al., 2025). Figure 4 shows a clear trend (across test benchmarks) of how we achieve the best performance once all the modifications are introduced. In particular, the most important change happens once the positional encodings are incorporated. `RoPE` is likely the main driver of this behavior, as it is making the model prioritize closer points to inform its predictions. However, in the remaining cases it is only feasible to achieve the desired behavior when combining all the modifications together, such as when learning ordered patterns.

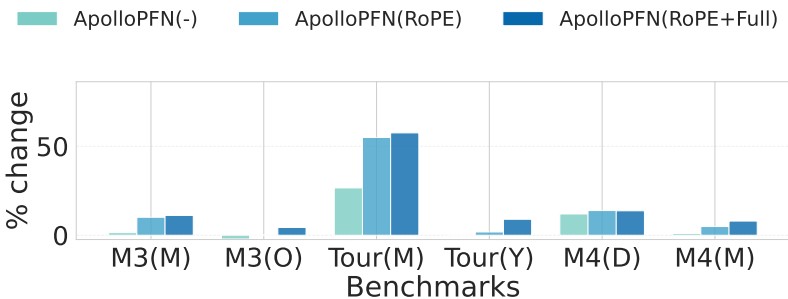

Figure 4: **Our interventions improve performance on time-series data.** Ablation on the use of RoPE and full attention. We compare the effect of progressively adding RoPE and full attention in several benchmarks against the baseline of `TabPFN-TS`.

## 5 EMPIRICAL EVALUATION

We now comprehensively compare `ApolloPFN` in several forecasting scenarios and against different forecasting models. Overall, `ApolloPFN` performs incredibly well on challenging time-series benchmarks that have exogenous information (Table 1 and Table 2). Furthermore, `ApolloPFN` has strong zero-short performance on classical benchmarks which do not contain exogenous information (Table 3), even against much larger models like Moirai-Large and Chronos-Large, which have $30 - 70\times$ more parameters than `ApolloPFN`, which only has 11M parameters.

### 5.1 ZERO-SHOT PERFORMANCE WITH EXOGENOUS FEATURES

Unfortunately, most publicly available time-series benchmarks in literature do not contain exogenous features (see GIFT-Eval (Aksu et al., 2024)) and we are restricted to a limited set such as the electricity price forecasting (Lago et al., 2021) or the M5 competition (Makridakis et al., 2022).

The electricity price forecasting dataset consist of hourly measurements of electric prices (Lago et al., 2021) for five major markets in Europe, namely Nord Pool (NP), PJM (COMED zone), France (FR), Belgium (BE), and Germany (DE). These datasets contain exogenous variables such as system load and power generation measurements. We provide a detailed description of the time spans and exogenous features for each market in Appendix C.2.

The M-series suite of benchmarks constitutes a comprehensive evaluation on how a model would perform across varying prediction lengths, different frequencies (hourly, daily, weekly, quarterly, yearly), and distinct sources of data, resulting in widely different time-series behaviors. It is worth mentioning that these M-series competitions: M1 (Makridakis & Hibon, 1979), M2 (Makridakis et al., 1993), M3 (Makridakis & Hibon, 2000), M4 (Makridakis et al., 2020) and, M5 (Makridakis et al., 2022) have been a consistent benchmark to evaluate forecasting models throughout the years. However, despite its breadth, only the M5 competition dataset (Makridakis et al., 2022) contains

| sCRPS | DE(24) | NP(24) | FR(24) | BE(24) | PJM(24) | DE(48) | NP(48) | FR(48) | BE(48) | PJM(48) |
|---|---|---|---|---|---|---|---|---|---|---|
| ApolloPFN[(0x)] | 0.040 | **0.038** | **0.040** | **0.042** | **0.040** | **0.056** | **0.053** | 0.069 | **0.058** | **0.057** |
| TabPFN-TS[(0x)] | **0.033** | 0.048 | 0.067 | 0.048 | 0.047 | 0.065 | 0.055 | **0.068** | 0.073 | 0.069 |
| Moirai-Large[(†x)] | 0.078 | 0.082 | 0.079 | 0.082 | 0.078 | 0.120 | 0.124 | 0.121 | 0.123 | 0.121 |
| Chronos-Large[(0)] | 0.119 | 0.110 | 0.139 | 0.117 | 0.107 | 0.088 | 0.106 | 0.105 | 0.089 | 0.094 |
| Sundial-Base[(0)] | 0.152 | 0.147 | 0.151 | 0.150 | 0.149 | 0.097 | 0.099 | 0.096 | 0.095 | 0.097 |

Table 1: **ApolloPFN beats other neural forecasters that leverage exogenous information.** sCRPS results on electric price forecasting across different datasets and prediction horizons (24, 48). [(0x)] denotes zero-shot forecasters that leverage exogenous information. [(†x)] denotes forecasters that leverage exogenous information but were exposed to the data during training. [(0)] denotes zero-shot univariate forecasters that do not use exogenous information. Best results for each dataset are **bold** and second best are underlined.

| Level | RMSSE | M5(D-B) | M5(W-B) | M5(M-B) | M5(D-S) | M5(W-S) | M5(M-S) |
|---|---|---|---|---|---|---|---|
| **State** | ApolloPFN[(0x)] | **0.580** | 1.652 | **2.191** | 0.973 | **1.561** | **2.588** |
| | TabPFN-TS[(0x)] | 0.608 | 1.253 | 2.580 | 1.006 | 1.666 | 2.636 |
| | Moirai-Large[(†x)] | 0.844 | 1.669 | 3.546 | 0.992 | 1.710 | 2.882 |
| | Chronos-Large[(0)] | 0.655 | **1.237** | 2.484 | 1.007 | 1.847 | 2.788 |
| | Sundial-Base[(0)] | 0.720 | 2.010 | 2.405 | **0.933** | 1.649 | 2.841 |
| **Store** | ApolloPFN[(0x)] | 0.675 | 1.829 | **2.208** | 0.990 | **1.449** | **2.049** |
| | TabPFN-TS[(0x)] | **0.651** | 1.729 | 2.278 | 1.024 | 1.572 | 2.119 |
| | Moirai-Large[(†x)] | 0.900 | 2.004 | 3.053 | 0.984 | 1.539 | 2.334 |
| | Chronos-Large[(0)] | 0.709 | **1.715** | 2.272 | 0.998 | 1.601 | 2.250 |
| | Sundial-Base[(0)] | 0.733 | 2.108 | 2.536 | **0.922** | 1.452 | 2.202 |

Table 2: RMSSE results on M5 at a state and store level for different data aggregations. We have brand level data (B) on the left and SKU level data (S) on the right for the following frequencies: Daily (D), Weekly (W), and Monthly (M). [(0x)] denotes zero-shot forecasters that leverage exogenous information. [(†x)] denotes forecasters that leverage exogenous information but were exposed to the data during training. [(0)] denotes zero-shot univariate forecasters that do not use exogenous information. Best results for each dataset-level are **bold**, and second best are underlined.

exogenous information such as price and promotional events to inform the predictions. The M5 dataset contains units sold daily for a given SKU (product) with identifying attributes such as brand, store and state. At the SKU and store level, M5 contains over 30K time-series. We create multiple versions of the M5 dataset by aggregating across time (to weekly and monthly grains) and across geographies (to state and store grains).

Tables 1 and 2 compare the `ApolloPFN` model against foundational forecasters that leverage exogenous information such as `TabPFN-TS` and `Moirai-Large`, and univariate foundational forecasters such as `Chronos-Large` and `Sundial-Base` against electricity forecasting and M5 aggregations benchmarks. In the electricity forecasting benchmark, ApolloPFN achieves on average 12% improvement over the next best model (`TabPFN-TS`), and achieves SOTA across most datasets. In the M5 aggregations benchmark, it achieves SOTA performance on most aggregation levels and remains highly competitive with much larger foundational models.

## 5.2 PERFORMANCE ON CLASSICAL UNIVARIATE BENCHMARKS

Given the limited availability of large-scale publicly accessible time-series datasets, most neural forecasting models in the literature utilize all or a substantial portion of the M-competition data for

| sCRPS | M1(M) | M1(Y) | M3(M) | M3(O) | M4(D) | M4(M) | M4(Y) | Tou(M) | Tou(Y) |
|---|---|---|---|---|---|---|---|---|---|
| ApolloPFN | 0.152 | 0.142 | 0.094 | **0.034** | **0.023** | **0.092** | 0.113 | **0.084** | 0.137 |
| TabPFN-TS | 0.169 | 0.123 | 0.106 | 0.035 | 0.027 | 0.096 | 0.115 | 0.203 | 0.146 |
| Moirai-Large | **0.135** | 0.210 | **0.093** | 0.035 | 0.033 | 0.117 | 0.187 | 0.275 | 0.275 |
| Chronos-Large | 0.173 | **0.119** | 0.113 | 0.036 | 0.028 | 0.108 | **0.106** | 0.155 | **0.103** |
| Sundial-Base | 0.157 | 0.183 | 0.121 | 0.047 | 0.026 | 0.116 | 0.160 | 0.126 | 0.174 |

Table 3: **ApolloPFN performance in classical univariate benchmarks.** Best results for each dataset are **bold**, and second best are underlined.

training. Consequently, this practice complicates a fair and unbiased comparison of zero-shot model performance on these benchmarks. In Table 3, we compare `ApolloPFN` against several of the best performing univariate foundational models. Most notably, `ApolloPFN` performs 10% better than `TabPFN-TS` on average and achieves SOTA across the different benchmarks.

## 6 CONCLUSION

`ApolloPFN` provides a time-series specific PFN model that gracefully accommodates exogenous variables, and achieves state-of-the-art zero-shot forecasting performance. The strong performance of this new PFN model is enabled through proposing architectural innovations, and a synthetic data generation process. It is notable that `ApolloPFN` can modulate the effect of different exogenous covariates on each time-series independently of each other. For example, if there is a product that does not respond to promotional events then `ApolloPFN` would not predict a lift for future promotional events, while other models might do if the majority of the products had a positive response during training.

Given the strong performance of `ApolloPFN`, it would be exciting to investigate further developments in the future. For example, the current reliance on standard quadratic attention prohibits applicability to very long series (>10K). It would also be enlightening to theoretically analyze the connection between the complexity of the synthetic data, and the performance and generality of the model. Moreover, it could be possible to further enhance the efficiency and time-series specific biases of the architecture through representing model parameters and attention with structured matrices (Potapczynski et al., 2024b).

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

## APPENDIX OUTLINE

The appendix is composed of the following sections

- Appendix A discusses the architectural details of `TabPFN` (Hollmann et al., 2023; 2025).
- Appendix B motivates and explains the different graph generation algorithms used during training.
- Appendix C elaborates on different details for the benchmark evaluations such as the evaluation metrics and the data sources.

## A   TABPFN ARCHITECTURE

In this section we elaborate on additional details that were not covered in Section 2.2. Assume that we have the following $N_{\text{train}}$ observations for our target $(y_i)_{i=1}^{N_{\text{train}}}$, $N = N_{\text{train}} + N_{\text{test}}$ observations for covariate information $(\boldsymbol{x}_i)_{i=1}^{N}$ where each $\boldsymbol{x}_i \in \mathbb{R}^{F'}$ and we want to make $N_{\text{test}}$ predictions for the target $(y_i)_{i=1}^{N_{\text{test}}}$.

The goal of the preprocessing step is to transform the information of $(\boldsymbol{x}_i)_{i=1}^{N}$ and $(y_i)_{i=1}^{N_{\text{train}}}$ into an embedding $\boldsymbol{Z} \in \mathbb{R}^{N \times F \times D}$ as used in Equation 1. In terms of the target, we first create a tensor $\tilde{\boldsymbol{Y}} \in \mathbb{R}^{N \times 2}$ by first z-scoring all the train targets, $\tilde{Y}_{i,1} = (y_i - \mu_{\text{train}})/\sigma_{\text{train}}$ where $\mu_{\text{train}} = \frac{1}{N_{\text{train}}} \sum_{i=1}^{N_{\text{train}}} y_i$ and $\sigma_{\text{train}}^2 = \frac{1}{N_{\text{train}}-1} \sum_{i=1}^{N_{\text{train}}} (y_i - \mu_{\text{train}})^2$ for the positions of $i = 1, \ldots, N_{\text{train}}$ and then by setting

the rest of the $N_{\text{test}}$ positions $i = N_{\text{train}} + 1, \ldots, N$ as $\tilde{Y}_{i,1} = \mu_{\text{train}}$. Then the other column of $\tilde{Y}$ would be filled with $\tilde{Y}_{i,2} = 0$ if the entry is observed ($i = 1, \ldots, N_{\text{train}}$) and $\tilde{Y}_{i,2} = -2$ if not ($i = N_{\text{train}} + 1, \ldots, N$). After than we create $Y \in \mathbb{R}^{N \times D}$ by embedding $\tilde{Y}$ with a linear layer on a $D$ dimensional space as $Y = \tilde{Y} W_Y$ where $W_Y \in \mathbb{R}^{2 \times D}$.

An analogous procedure is done for each of the features in $x_i \in \mathbb{R}^{F'}$ after first grouping them in pairs as discussed in Hollmann et al. (2025). The grouping can done easily with a reshape as follows. If we have $\tilde{X}'_i = x_i$, then $\tilde{X} = \text{Reshape}(\tilde{X}', (N, F'/2, 2))$ would have the desired effect (assuming that $F'$ is divisible by 2, else we 0 pad the feature dimension). After z-scoring each of the $f = 1, \ldots, F'/2$ features we then compute $X = \tilde{X} W_X \in \mathbb{R}^{N \times F-1 \times D}$ where $W_X \in \mathbb{R}^{2 \times D}$ and $F = F'/2 + 1$. After the embedding $X$ is constructed we then add a fixed random positional encoding $\Omega \in \mathbb{R}^{F-1 \times D}$ to each feature shared across all $N$ samples. In other words we do $X_i \leftarrow X_i + \Omega$ for all $i = 1, \ldots, N$. Finally, we set $Z = [X, Y] \in \mathbb{R}^{N \times F \times D}$ which would then be the embedding pass to the architecture seen in Figure 5 and discussed in Section 2.2 Equation 1.

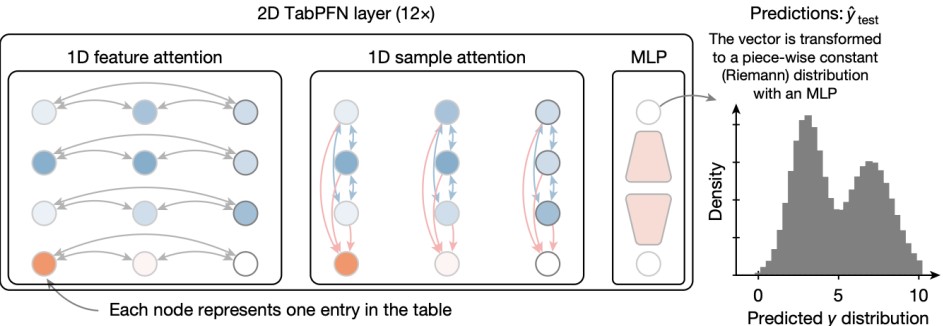

Figure 5: **How TabPFN combines attention across features and samples.** Taken from Hollmann et al. (2025), the figure illustrates the main components of the `TabPFN` architecture discussed in Equation 1 plus the translation of the embedding into a Riemann distribution approximation of the PPD $p(y_{\text{test}}|x_{\text{test}}, \mathcal{D}_{\text{train}})$.

The transformation of $Z \in \mathbb{R}^{N \times F \times D}$ into the Riemman approximation of the PPD is done with another linear layer as $Z_{N_{\text{train}}:, -1, :} W_Z \in \mathbb{R}^{N_{\text{test}} \times Q}$ where $W_Z \in \mathbb{R}^{D \times Q}$ and $Q$ is the number of quantiles needed to compute the PPD.

## B  DATA GENERATION

### B.1  GRAPH ALGORITHMS

As explained in Section 2.2 and Section 4.1 we need to randomly generate graphs (DAGs) to define diverse SCMs for our synthetic data procedure. The initial procedure to construct a graph (Hollmann et al., 2023) was through a MLP, where each node is connected to all other nodes in the next layer and the depth of the MLP is the depth of the graph which culminates with 1 node at the end which would be the target. To illustrate, if we have a 3-layered MLP with a width of 10 then we would have a graph with $21 = 10 + 10 + 1$ nodes and $110 = 10 \times 10 + 10 \times 1$ edges (assuming that the MLP is fully connected). A step to reduce the density of the graph is to drop some edges uniformly at random or by blocks (Hollmann et al., 2023).

In Hollmann et al. (2025), the authors adopted a "more realistic" DAG generation by using a classical algorithm in the study of random networks called the random growing network with redirection (Krapivsky & Redner, 2023) which is represented in Algorithm 1.

As seen in Figure 6 (*Top*), a characteristic of Algorithm 1 is that it generates graphs with many root nodes (as each added root node in might never get an incoming edge) and, if the redirection probability $\rho$ is high then several of the root nodes might point to the first node. When selecting which features to use from a graph the root nodes are always excluded (Hollmann et al., 2025) and so having a graph that has many root nodes is not necessarily optimal. Furthermore, if the graph happens to concentrate in a few nodes, then many of the features would not be related (that is, there

---

**Algorithm 1** Random Growing Network with Redirection and Preferential Attachment

---

**Require:** $V$: total number of nodes, $\rho$ redirection probability
1: Initialize graph $G$ with nodes $n = 0$, $n = 1$ and edge $(1, 0)$
2: Initialize in-degree $k_j = 0$ for all $j \neq 0$, $k_0 = 1$
3: **for** $n = 2, \ldots, V - 1$ **do**
4:     Compute attachment probabilities for all nodes $i < n$
5:     $\Pi_i = \frac{k_i + 1}{\sum_{j=0}^{n-1}(k_j + 1)}$
6:     Select target node $t$ with probability $\Pi_t$
7:     Sample $u \sim U(0, 1)$
8:     **if** $u < \rho$ **then**
9:         Connect with target, add edge $(n, t)$
10:        Update: $k_t \leftarrow k_t + 1$
11:     **else**
12:        Connect with target's only descendant, add edge $(n, d)$
13:        Update: $k_d \leftarrow k_d + 1$
14:     **end if**
15: **end for**
16: **return** DAG $G = (V, E)$

---

would not be a path that connects them) making many of the features in the dataset not informative about the target.

---

**Algorithm 2** Single Root Node Random Growing Network

---

**Require:** $V$: total number of nodes, $\rho$ additional attachment probability
1: Initialize graph $G$ with nodes $n = 0$, $n = 1$ and edge $(1, 0)$
2: Initialize in-degree $k_j = 0$ for all $j \neq 0$, $k_0 = 1$
3: **for** $n = 2, \ldots, V - 1$ **do**
4:     Compute attachment probabilities for all nodes $i < n$
5:     $\Pi_i = \frac{k_i + 1}{\sum_{j=0}^{n-1}(k_j + 1)}$
6:     Select target node $t$ with probability $\Pi_t$
7:     Select an additional source node uniformly at random from $s \in \{0, \ldots, n - 1\} \setminus \{t\}$
8:     Source node connects to new node, add edge $(s, n)$
9:     Update: $k_n \leftarrow k_n + 1$
10:    Sample $u \sim U(0, 1)$
11:    **if** $u < \rho$ **then**
12:       Target connects to new node, add edge $(t, n)$
13:       Update: $k_n \leftarrow k_n + 1$
14:    **end if**
15: **end for**
16: Eliminate cycles in $G$ (if any)
17: **return** DAG $G = (V, E)$

---

To generate our graphs to train `ApolloPFN`, we essentially reverse the mechanisms of Algorithm 1 that makes the output graphs have several nodes and unconnected features. That is, we always incorporate nodes in a graph by having a prior node connect to it, and also, make it connect to a popular node with probability $\rho$. All the steps are in Algorithm 2 and we can see in Figure 6 (*Bottom*) how we generate graphs that are connected via some path and that only have one single root node by construction. We show that generating data using Algorithm 2 accelerates training as seen in Figure 3. Similar to Hollmann et al. (2025) we sample the number of total nodes as $\log V \sim \mathcal{U}[a, b]$ but we sample $\rho \sim \mathcal{B}(\alpha, \beta)$ using a Beta distribution instead of the Truncated Gamma distribution used in Hollmann et al. (2025).

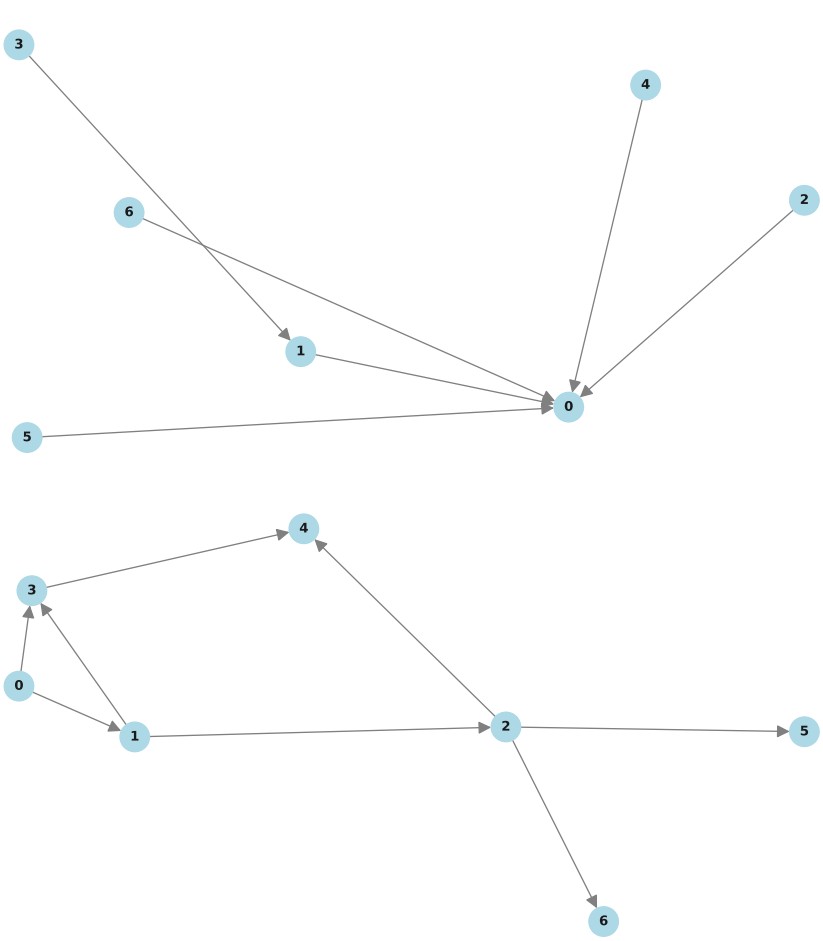

Figure 6: **Example graphs from distinct graph generation algorithms**. *(Top)* Example graph sampled via growing random networks with redirection and preferential attachment (Krapivsky & Redner, 2023). *(Bottom)* Example graph using our single root node growing random network.

## C  EXPERIMENTAL DETAILS

### C.1  EVALUATION METRICS

In this section we document the evaluation metrics for our experiments.

Given $\alpha_1 < \cdots < \alpha_Q$ quantiles, with $\alpha_j \in (0, 1)$ sCRPS is defined as:

$$\text{sCRPS}(y, \hat{y}) = \frac{\sum_{t=T+1}^{T+H} \frac{2}{Q} \sum_{j=1}^{Q} \alpha_j \left(y_t - \hat{y}_t^{\alpha_j}\right)_+ + (1 - \alpha_j) \left(y_t - \hat{y}_t^{\alpha_j}\right)_-}{\sum_{t=T+1}^{T+H} |y_t|}$$

where $(\cdot)_+$ is the positive part and $(\cdot)_-$ the negative part functions. Additionally, $\hat{y}_t^{\alpha_j}$ represents the $\alpha_j$-th quantile prediction for time step $t$. The sCRPS captures how well our accurate are our probabilistic predictions but we scale them by the norm of the values of the SKU to weight all SKUs equally.

To evaluate M5, we used the suggested RMSSE metric from Makridakis et al. (2022). This metric is defined as:

$$\text{RMSSE}(y, \hat{y}) = \frac{\frac{1}{H} \sum_{t=T+1}^{T+H} (y_t - \hat{y}_t)^2}{\frac{1}{T-1} \sum_{t=2}^{T} (y_t - y_{t-1})^2}$$

The motivation for RMSSE is three-fold. First, it compares the predictions against a naive one baseline, giving us a sense of how easy or hard it is to make predictions for this SKU. Second, it down weights SKUs that might have not many sales in the beginning periods similar to the case in Figure 2 (c). Third, it focuses on a square error with penalizes models that do not capture spikes in behavior.

### C.2  DATA

All the dataset that we used are publicly available and can be found either the `GiftEval` (Aksu et al., 2024) repository or the `LOTSA` (Woo et al., 2024) huggingface repository.

Below we have a Table 4 with the dataset and citations for reference

| Dataset | Source |
|---------|--------|
| M1 | Makridakis & Hibon (1979) |
| M3 | Makridakis & Hibon (2000) |
| M4 | Makridakis et al. (2020) |
| Tourism | Hyndman et al. (2008) |
| M5 | Makridakis et al. (2022) |
| Electric Price | Lago et al. (2021) |

Table 4: Data sources used for benchmarking.

In terms of electric prices (Lago et al., 2021), we have: the Nord pool (NP) market which is one of the largest European power markets containing hourly measurements from 2023-01-01 to 2018-12-24. The NP dataset comes with exogenous variables measuring the grid load and wind power. We then have the zonal prices for the COMED area of Pennsylvania, New Jersey and Maryland (PJM) containing hourly measurements from 2023-01-01 to 2018-12-14. The PJM dataset comes with exogenous measurements of the system load and zonal load. Next, we have the French electricity market (FR) containing hourly measurements from 2011-01-09 to 2016-12-31. The FR dataset contains exogenous measurements of system load and power generation. Then, we have the Belgian electricity market (BE) containing hourly measurements from 2011-01-09 to 2016-12-31. The BE dataset contains exogenous measurements of system load and power generation. Finally, we have the German electricity market (DE) containing hourly measurements from 2012-01-09 to 2017-12-31. The DE dataset contains exogenous measurements of zonal load and both solar and wind generation measurements.

| sCRPS | ApolloPFN | TabPFN-TS |
|---|---|---|
| Favorita(S) | 0.073 | 0.081 |
| Favorita(C) | 0.075 | 0.099 |
| Favorita(St) | 0.095 | 0.105 |

Table 5: sCRPS results on weekly Favorita at the state (S), city (C) and store (St) level.

| Level | RMSSE | M5(D-B) | M5(W-B) | M5(M-B) | M5(D-S) | M5(W-S) | M5(M-S) |
|---|---|---|---|---|---|---|---|
| **State** | ApolloPFN | **0.580** | **1.652** | 2.191 | **0.973** | **1.561** | **2.588** |
| | TabPFN-TS | 0.651 | 1.729 | 2.278 | 1.024 | 1.572 | 2.119 |
| | SimpleApolloPFN | 1.358 | 4.650 | **1.667** | 1.042 | 5.058 | 3.443 |
| **Store** | ApolloPFN | 0.675 | 1.829 | 2.208 | **0.990** | **1.449** | **2.049** |
| | TabPFN-TS | **0.651** | **1.729** | 2.278 | 1.024 | 1.572 | 2.119 |
| | SimpleApolloPFN | 1.201 | 7.20 | **1.827** | 1.355 | 3.901 | 2.977 |

Table 6: RMSSE results on M5 at a state and store level for different data aggregations. We have brand level data (B) on the left and SKU level data (S) on the right for the following frequencies: Daily (D), Weekly (W), and Monthly (M). SimpleApolloPFN is our PFN method trained with no SCMs but rather simple exogenous interventions like promotional spikes or decreases and upward or downard phase shifts in the time series.

# D    HYPERPARAMETER DETAILS

## D.1    SCM GENERATION

The sampling procedure for our SCMs is the following. We selected the number of nodes uniformly from a minimum of 20 to a maximum of 150. Each node then contains a state of dimensionality 6 which we propagate through the graph. Moreover, when using a MLP edge we select our activation from the following options: tanh, sine, abs, identity, log, sigmoid, smooth relu, modulo and step wise (or indicator). The entries weights of the layers in the MLPs are sampled from $\mathcal{N}(0, 1)$. The sample frequencies $\phi$ are sampled from $\log \phi \sim \mathcal{U}(1, 10)$ and the amplitudes $\alpha \sim \mathcal{N}(0, 1)$.

## D.2    TRAINING

We train our models for 300K steps using a batch size of 64 with a learning rate of 1e-4, no weight decay, 20K linear warm-up steps and we used a cosine annealing schedule that terminates with a learning rate of 1e-6. We vary the number of samples and number of features available to the model for each batch. The number of samples ranges from 34 to 512 and the number of features from 2 to 64 and we predict for a horizon of up to 128.

# E    ADDITIONAL ABLATIONS

This section contains several experiments. The performance comparison of weekly Favorita across different geographical aggregations 5. Favorita is a grocery demand forecasting task with data from the ecuadorian Corporación Favorita (Favorita). This datasets consist of weekly unit demand across several products with indicators of promotional activity that we use as exogenous information. The dataset can be found here `https://www.kaggle.com/c/favorita-grocery-sales-forecasting`.

In Table 8 we showed the effect of adding causal masking to the attention mechanism. As we can see, forcing the model to only look backward imposes a performance limitation into it. In contrast, allowing the model to simultaneously make predictions by considering the influence of predictions ahead aid in performance.

| sCRPS | M1(M) | M1(Q) | M1(Y) | M3(M) | M3(O) | M3(Q) | M3(Y) | M4(D) | M4(M) | Tour(M) | Tour(Q) | Tour(Y) | AVG |
|---|---|---|---|---|---|---|---|---|---|---|---|---|---|
| None | 0.178 | 0.100 | 0.113 | 0.101 | 0.035 | 0.078 | 0.135 | 0.024 | 0.098 | 0.168 | 0.112 | 0.121 | 0.105 |
| Sine | 0.169 | 0.089 | 0.125 | 0.101 | 0.035 | 0.077 | 0.132 | 0.076 | 0.099 | 0.163 | 0.098 | 0.123 | 0.107 |
| Learnt | 0.177 | 0.106 | 0.121 | 0.100 | 0.034 | 0.076 | 0.136 | 0.023 | 0.096 | 0.130 | 0.090 | 0.159 | 0.104 |
| RoPE | 0.151 | 0.086 | 0.146 | 0.093 | 0.034 | 0.068 | 0.130 | 0.022 | 0.091 | 0.085 | 0.070 | 0.147 | 0.094 |

Table 7: RoPE embeddings is the best performing positional encoding strategy across different benchmarks.

| sCRPS | M1(M) | M1(Q) | M1(Y) | M3(M) | M3(O) | M3(Q) | M3(Y) | M4(D) | M4(M) | Tour(M) | Tour(Q) | Tour(Y) | AVG |
|---|---|---|---|---|---|---|---|---|---|---|---|---|---|
| Causal | 0.346 | 0.123 | 0.127 | 0.212 | 0.096 | 0.149 | 0.209 | 0.097 | 0.246 | 0.346 | 0.192 | 0.155 | 0.191 |
| Non | 0.151 | 0.086 | 0.146 | 0.093 | 0.034 | 0.068 | 0.130 | 0.022 | 0.091 | 0.085 | 0.070 | 0.147 | 0.094 |

Table 8: Adding causal masking (Causal) into the architecutre significantly decreases performance compare to not using it (Non).

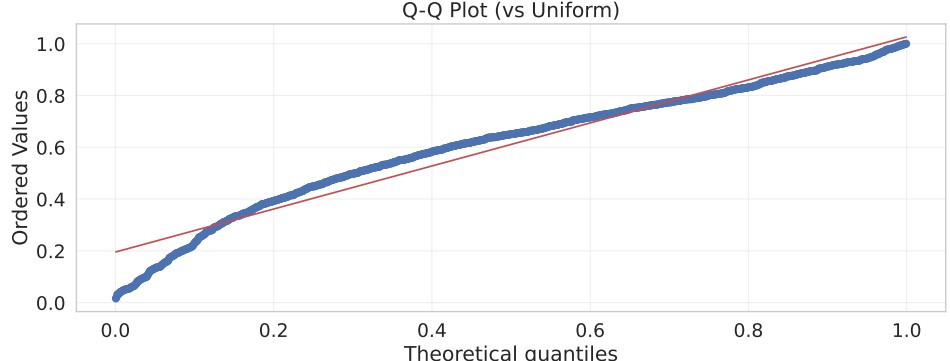

Figure 7: Q-Q plot comparing ApolloPFN's CDF over the true targets against a $\mathcal{U}[0,1]$ distribution. Most of the quantiles are well calibrated except the lower ones.

