# OpenReview forum: "Time-Aware Prior Fitted Networks for Zero-Shot Forecasting with Exogenous Variables"
_ICLR.cc/2026/Conference — Submitted to ICLR 2026_

### Official Review · Reviewer_hrkC · 2025-10-26

**Soundness:** 2
**Presentation:** 2
**Contribution:** 2
**Rating:** 4
**Confidence:** 4

**Summary:**

This paper introduces ApolloPFN, a Prior-Fitted Network (PFN) designed for zero-shot time series forecasting that incorporates exogenous covariates.

**Strengths:**

1. The focus on zero-shot forecasting with exogenous variables addresses real industrial needs where fine-tuning is undesirable.
2. The evaluation covers multiple realistic benchmarks, and shows consistent improvements especially on tasks with exogenous information.

**Weaknesses:**

1. While the combination is effective, individual components are relatively incremental. RoPE is a standard positional encoding, and the SNGN algorithm is essentially a reversal of existing graph generation methods. The main contribution seems to be recognizing that tabular PFNs need temporal adaptations, which is somewhat obvious in hindsight.
2. How are the frequencies (φ₁, φ₂) and amplitudes (α₁, α₂) sampled for time-dependent root nodes? This is critical for understanding what temporal patterns the model learns.
3. The paper doesn't discuss how synthetic SCMs relate to real time series structures. Why should random DAGs with MLPs/decision trees as node functions capture real forecasting scenarios?
4. What's the distribution over graph sizes, number of features, etc.. These details matter for reproducibility.
5. The paper identifies that TabPFN-TS "fails to understand temporal autocorrelations" but doesn't explain why adding a running index feature isn't sufficient to break order invariance.
6. How does ApolloPFN perform when the forecast horizon H is much larger than what was seen during training?

**Questions:**

See weaknesses.

---

> ### Author Response · Authors · 2025-11-22
>
> Thank you for your review. We would like to encourage you to see our work through its impact (we surpass SOTA 3-10% Tables 1-3) and not its complexity. To this point, we see the simplicity of our architectural modifications as a strength and not a weakness. Our changes facilitate adoption and highlight the minimal set of modifications that yield a consistent positive impact as illustrated in Figure 4. In the end, I think you would agree that we would rather have a simple method that consistently yields better results rather than a complex method that yields marginal improvements. To this end, we have added a series of ablations studying the impact of different positional encodings as seen in Table 7 in the new appendix E.
>
> There is also a misunderstanding about our main contribution. Indeed it is trivial to argue that a PFN trained on synthetic time-series data would perform better on time-series data. There is already prior work for univariate time series [1,2] that illustrates this point. The question is _how_ do we create synthetic time-series data with exogenous information that will perform in real-life datasets! Note that for this scenario, we have to create synthetic time-series data with multiple covariates impacting its behavior and although it is easy to come up with a synthetic process (say add random spikes given a categorical covariate) it is not guaranteed that that process would actually perform well in real data. Indeed, based on your feedback, we added to the new appendix E Table 6 a series of ablations showing how more trivial exogenous synthetic processes do not perform well in benchmarks. These exogenous processes are promotional spikes and phase shifts. Please note how in some datasets like M5(W-B) the performance is over 4.5x worse,
>
> You raise an interesting question that we want to explore in future work, that is, why do synthetic SCMs work so well in practice when the generating process seems to be distant from the actual downstream application. In contrast to your point, we are actually not sure that synthetic SCMs should capture real forecasting scenarios to work well in practice. In this respect your hypothesis seems to imply that only synthetic data that interpolates / approximates the downstream application is valuable. Our experience suggests that this is not the case. Our model makes predictions in-context and we believe that once it has seen sufficiently hard examples it learns some favorable mechanism to make predictions that actually works really well in practice. Our experimental results bolster our perspective.
>
> We have added a new appendix D answering your hyperparameter questions. In particular we ran graphs of sizes ranging from 20 to 150 nodes.
>
> Finally, we agree with your point that adding a running index is sufficient information for TabPFN-TS to break the order invariance. We were also initially puzzled by this fact. We found that the attention mechanism of TabPFN-TS does not have a notion of locality as seen in Figure 2 (c). This finding is what motivated us to employ RoPE embeddings which, as also seen in Figure (c), makes our time-series model work as expected. In terms of the horizon, we used a maximum of 128 and tested within this range (similar to TabPFN-TS). If a higher horizon is needed we do suggest training a model for that case but longer horizons will be limited by the GPU memory.
>
> We would greatly appreciate if you could reconsider your score given our responses and our additional experiments.
>
> _References_
>
> [1] Dooley et al. 2023. ForecastPFN: Synthetically-Trained Zero-Shot Forecasting.
>
> [2] Bhethanabhotla et al. 2024. Mamba4Cast: Efficient Zero-Shot Time Series Forecasting with State Space Models.

---

### Official Review · Reviewer_pg4u · 2025-10-28

**Soundness:** 2
**Presentation:** 1
**Contribution:** 2
**Rating:** 2
**Confidence:** 4

**Summary:**

This paper proposes ApolloPFN, a time-aware prior-fitted network for zero-shot time-series forecasting with exogenous variables. Existing foundation models (e.g., Chronos, Moirai) often ignore exogenous covariates or require fine-tuning, while TabPFN-TS lacks temporal inductive bias due to its i.i.d. training assumption. ApolloPFN addresses these issues through two innovations: (1) a temporal synthetic data generation procedure and (2) architectural modifications adding RoPE positional encodings and expanded attention to capture temporal dependencies. Experiments on M5 and electricity price forecasting benchmarks demonstrate that ApolloPFN achieves state-of-the-art zero-shot performance with exogenous inputs and remains competitive on classical univariate benchmarks despite being smaller than large TSFMs.

**Strengths:**

1. The proposed temporal SCM generation procedure and the SNGN algorithm are well-motivated and supported by empirical evidence. The design effectively introduces temporal dependencies into the synthetic data, which allows the model to learn meaningful temporal structures during training. This approach represents a thoughtful and principled way to bridge the gap between tabular PFNs and time-series forecasting tasks.

**Weaknesses:**

1. Many of the “failure mode” demonstrations (e.g., Fig. 2) rely on single illustrative examples rather than aggregated or statistically supported analyses. Without quantitative evidence across a larger number of series or benchmarks, it is difficult to assess whether these issues with TabPFN-TS are systematic or merely anecdotal, which somewhat weakens the empirical foundation of the argument.

2. Most datasets containing exogenous covariates (such as M5 and electricity price forecasting) are relatively small in scale and limited in diversity. It remains unclear how ApolloPFN would generalize to larger, more complex multivariate datasets.

3. The experimental comparison includes a limited set of baselines and datasets, which makes the empirical results less solid and comprehensive.

4. The proposed architectural modifications—such as the incorporation of positional embeddings and expanded attention—are relatively standard and have been widely adopted in time-series Transformer variants. As a result, the architectural novelty of ApolloPFN is somewhat limited, with the main contribution lying more in the adaptation of PFNs to temporal contexts rather than in introducing fundamentally new modeling mechanisms.

5. The overall writing and exposition of the paper are somewhat average. Several sections could benefit from clearer explanations, better structural flow, and more precise terminology. Improving the presentation would help readers more easily grasp the key motivations, methodological details, and experimental setups.

**Questions:**

1. Since the model employs an encoder-only Transformer architecture, is there a defined maximum sequence length for T (time steps) and F (features) during training? If such limits exist, what are the specific maximum values for T and F, and what are the implications of these limits on the model’s ability to process long or high-dimensional time series? Additionally, are there strategies, such as memory-efficient attention mechanisms that could potentially allow the model to handle sequences beyond these limits?

2. Is the model capable of handling variable-length sequences for both T (time steps) and F (features) during inference? If so, was this flexibility explicitly supported during training, for example by including samples of varying lengths, to improve the model’s generalization across sequences of different sizes?

3. In the design of the “Expanding Attention” mechanism, what is the rationale behind allowing all points to attend to each other? Could a more restricted attention mechanism, such as causal or masked attention that only considers past or neighboring points, be sufficient to capture temporal dependencies?

4. What is the total amount and detailed configuration of the synthetic data used for training? For example, how many samples, series lengths, or graph instances were generated?  And how significant is the impact of the synthetic data generation parameters on the final performance of the model?

---

> ### Author Response · Authors · 2025-11-22
>
> Thank you for your encouragement highlighting that our work is a thoughtful and principled approach to improve PFNs for time-series. We now address your feedback.
>
> One of the key objectives of our paper is to encourage the neural forecasting community to focus on forecasting problems that leverage exogenous information. Unfortunately, as we discussed in section 5.1, there are just a handful of benchmarks that meet this criteria. In this respect, we believe that you are penalizing us for the lack of benchmarks that are currently available. Indeed supporting our work is a way to highlight the relevance of this topic and encourage the community to create more benchmarks. Although we all agree that it would be ideal to have more exogenous benchmarks, we would like to clarify that M5 is not small scale. It contains 10 points of sales across 3K products at a daily frequency. This amounts to 30K individual time series with a wide range of behaviors. Moreover, as explained on section 5.1, we created an additional set of 10K time series by aggregating through different periods: weekly and monthly as well as different states and brands. In other words, this M5 dataset contains time series such as the erratic daily behavior of an unpopular product as well as highly seasonal patterns of a brand's sales at a national level. Finally, we also added results on Favorita in Table 5 in the new Appendix E which is another dataset that we found with exogenous information but that, unfortunately, is also similar in nature to M5.
>
> We would also like to encourage you to see our work through its impact (we surpass SOTA 3-10% Tables 1-3) and not its complexity. To this point, we see the simplicity of our architectural modifications as a strength and not a weakness. Our changes facilitate adoption and highlight the minimal set of modifications that yield a consistent positive impact as illustrated in Figure 4. In the end, I think you would agree that we would rather have a simple method that consistently yields better results rather than a complex method that yields marginal improvements.
>
> You raise a relevant critique that the illustrative examples of Section 3 Figure 2 might not highlight systematic errors. Fortunately, these systematic errors are quantified in the performance metrics in section 5. Indeed pitfalls like lack of recency bias or uncalibrated CIs which are specifically measured by RMSSE and sCRPs (appendix D).
>
> There is a maximum number of steps and number of features that we used for training, namely 512 steps and 64 features. These particular choices were taken to ensure that we would not run out of memory during training. To your point, we indeed expect a decline of performance if we were to go much further from either axis of number of steps or numbers of features. However, as seen in our experimental results in section 5, these maximums are sufficient for the model to be practically useful as we can tackle all benchmarks without a problem. In terms of memory-efficient attention mechanisms, we do want to explore in the future all the potential modifications that we can introduce to the attention mechanism to reduce its quadratic cost. Currently our modifications have focused on making the model improve performance but reducing cost is also relevant in practice.
>
> Indeed one of the most important aspects of our model is that it can handle variable T and F. During training we sample a distinct T and a F and then create a batch during training (see the new Appendix D for additional details). Actually all the benchmarks from Table 1 to Table 3 have different T (even within the same dataset) and different F so the generalization across different T and Fs is guaranteed through training.
>
> The rationale for expanding attention is that, when making predictions we should use all the information available to us. That is, if we are to make a prediction about the demand this week our model should also consider that next week there is a promotional event which could reduce the demand as people will pause their purchases until the event. If we were to impose a causal mask, then our model could not modify its predictions in response to an incoming promotional event. We added an ablation in Table 8 in the new appendix E showing that causal attention is quite damaging for performance.

---

> > ### Author Response · Authors · 2025-11-22
> >
> > We trained our models for ~300K steps each containing a batch of 64 sampled datasets totalling 19.2 M distinct datasets (distinct graphs) generated through training. The sample lengths varied from 34 to 512 and the number of features from 2 to 64 where a new sampled number was selected at each step. Overall, we saw that the most sensitive number that directly impacted performance was the number of sampled nodes on the graph where we set the minimum to be 20 nodes and the maximum 150. Particularly decreasing the maximum yielded notable decreases in performance. The rest of the hyperparameter choices yielded similar results. Please take a look at Appendix D.
> >
> > We would greatly appreciate that given our arguments and additional experiments you would reconsider raising your score.

---

> > > ### Comment · Reviewer_pg4u · 2025-11-27
> > >
> > > Thank you for your reply. However, real-world impact is only one aspect of evaluating a paper, and the experiments in this work are not particularly comprehensive. Of course, as the authors mentioned, there are few available datasets, but I believe this is not an unsolvable problem. Recently, many datasets have been publicly released, such as those from GIFT Eval. Factors like method scalability, novelty, and the writing quality of the paper are all part of the overall assessment. Overall, I tend to maintain my score while also considering the opinions of the other reviewers.

---

> > > > ### Author Response · Authors · 2025-11-28
> > > >
> > > > We greatly appreciate the engagement. Let's us further make some arguments to hopefully change your perspective.
> > > >
> > > > Unfortunately, GIFT-Eval does not have a single dataset with exogenous information. We would find contradictory to the points that we raise in our paper about how the community should stop solely prioritizing non-exogenous benchmarking (as GIFT-Eval) to then focus most of our experiments on these types of benchmarks. We are worried about not being able to break this equilibrium where the majority of the current neural forecasting research focuses on non-exogenous benchmarks while industry cares mostly about exogenous. Furthermore, the main appeal of our model is that it can handle any number of exogenous variables in a zero-shot fashion and we find it also contradictory to then prioritize the experiments of our model outside of this setting.
> > > >
> > > > The scalability of our technique is inherent to the use of attention and is the same limitation that many current neural forecasting models face. Scalability can therefore be dealt with with similar techniques as patching. However, in our industry use of our model and in the M5 benchmark with 30K time series we were still able to perform inference without a problem in a A100.
> > > >
> > > > Again, we appreciate the engagement. Please let us know if you would like us to elaborate further or tackle any other point that you would like us to address.

---

### Official Review · Reviewer_iujA · 2025-10-31

**Soundness:** 2
**Presentation:** 3
**Contribution:** 2
**Rating:** 4
**Confidence:** 3

**Summary:**

This paper describes a novel prior-data fitted network, ApolloPFN, that improves the drawbacks of TabPFN by 	adapting to the current date, synthetic data generation and time-aware architectural modifications. The model is then validated across datasets with exogenous  variables and shows improved inferences.

**Strengths:**

1. Paper is clear and well presented.
2. Framework reforms well across the two datasets considered by the authors (M5 dataset and electricity price forecasting).
3. Paper presents an algorithm that improves PFM architectures for exogenous features.

**Weaknesses:**

1. While the framework is established for multivariate datasets, the comparison with just two datasets seems limited. Other multivariate datasets like wind power forecasting etc can be used to understand the performance of this algorithm further.
2. Benchmark models can be improved by considering TTM and/or Flowstate which also works with exogenous features and is in the top 10 of Gift-Eval dataset.

**Questions:**

1. How well does the algorithm work with univariate datasets? Has it been tested with all the standard benchmark datasets like ETTh1 etc from time series foundation models literature.
2. How does the computational cost change of ApolloPFN as compared to TabPFN?

---

> ### Author Response · Authors · 2025-11-12
> **Reference for suggested benchmark**
>
> Thank you for your thoughtful review! We're working hard to make updates based on your comments and we will provide a comprehensive response in the following days. We would be grateful if you could please point us to the wind power forecasting dataset that you are suggesting in order to benchmark against it. To clarify, we are looking for datasets with exogenous information not multivariate datasets. Many thanks.

---

> > ### Author Response · Authors · 2025-11-22
> >
> > Thank you for your review and for acknowledging that our method shows improved inference results. One of the key objectives of our paper is to encourage the neural forecasting community to focus on forecasting problems that leverage exogenous information. Unfortunately, as we discussed in section 5.1, there are just a handful of benchmarks that meet this criteria. In this respect, we believe that you are penalizing us for the lack of benchmarks that are currently available. Indeed supporting our work is a way to highlight the relevance of this topic and encourage the community to create more benchmarks. We also added results on Favorita in Table 6 in the new Appendix E which is another dataset that we found with exogenous information but that, unfortunately, is also similar in nature to M5. We would like to clarify that TTM and Flowstate are models that handle multivariate time series, not exogenous covariates. Multivariate benchmarks tackle the task of making predictions of multiple targets which are all endogenous to a system such as humidity and temperature. But exogenous covariates are features that influence a target but that are not jointly forecasted.
> >
> > Our performance on univariate datasets is distinctly better than SOTA as seen in Table 3. Overall, we beat TabPFN-TS by about 10% and we used the canonical univariate benchmarks. You mentioned ETTh1 which is a benchmark commonly used for “long-context” prediction of Transformer models where there are specialized architectures that handle that context through techniques like patching. In our work we do not tackle long-context prediction, and our models are trained to handle up to a context of size 512. Indeed we believe there is an opportunity to apply techniques like patching to our work in order to deal with “long-context” benchmarks in future work. Finally, there is almost no additional computational cost when running ApolloPFN compared to TabPFN-TS as we overall have similar size models and the use of RoPE embeddings adds a small amount of overhead.
> >
> > In light of our response which addresses your feedback we would like to ask you if you could reconsider raising your score.

---

> ### Comment · Reviewer_iujA · 2025-11-24
>
> Thank you for the clarification. I wanted to check the claim about exogenous variables and TTM applicability. References like https://www.sktime.net/en/stable/api_reference/auto_generated/sktime.forecasting.ttm.TinyTimeMixerForecaster.html seem to suggest that TTM supports exogenous variables. I might be missing a point here, so can you please clarify further?

---

> > ### Author Response · Authors · 2025-11-28
> >
> > We greatly appreciate the engagement, let us clarify further!
> >
> > Several models support exogenous variables through training or fine-tuning but not in a zero-shot fashion as it is our main goal in the paper. For example, through the use of an additional MLP module, the popular NBEATs model can be extended to use exogenous variables [1] or, in general, any forecasting model can have a general module that gets trained to use the exogenous information as in [2]. But all this additional models need to be trained with the exogenous information.
> >
> > In the documentation that you shared , you can see that the `fit_strategy=full` for exogenous information which means that all the model parameters are going to be changed. Then you can see how there is a call to `forecaster.fit()` passing the exogenous information `X_train` before making predictions.
> >
> > Please let us know if we can provide any further clarifications.
> >
> > _References_
> >
> > [1] Olivares et al. 2013. Neural basis expansion analysis with exogenous variables: Forecasting electricity prices with NBEATSx
> >
> > [2] Potapczynski et al. Effectively Leveraging Exogenous Information across Neural Forecasters.

---

### Official Review · Reviewer_c4ks · 2025-10-31

**Soundness:** 3
**Presentation:** 2
**Contribution:** 2
**Rating:** 6
**Confidence:** 3

**Summary:**

The paper targets zero-shot forecasting with exogenous variables—a gap in many TSFMs (Chronos, Sundial, TimesFM, TimeLLM, LagLlama) that often ignore or require finetuning for exogenous covariates. It critiques TabPFN-TS (order-invariant tabular PFN with TS features added) and proposes ApolloPFN, contributing: (i) a time-aware synthetic data generator (graph-based with time-dependent roots) to better match TS priors; and (ii) architectural biases that respect temporal order. On benchmarks with exogenous info (e.g., M5 aggregations and electricity price), ApolloPFN achieves SOTA zero-shot performance and outperforms TabPFN-TS on average; it is competitive against Moirai and large univariate TSFMs.

**Strengths:**

Adapting the PFN paradigm to time series with native exogenous handling and explicit time-aware inductive bias is a meaningful advance. The critique of using order-invariant tabular FMs for TS is convincing (forecasting requires order sensitivity), and the synthetic-prior design tailored to TS is a good fit for PFN training. Given the practical importance of zero-shot + exogenous, the contribution is significant.

* The paper clearly articulates failure modes of TabPFN-TS (order-invariance, weak trend extrapolation, poor calibration under regime changes) and motivates architectural/time-aware changes. Examples illustrate the gap.


* The experimental comparisons show gains on electricity and M5 aggregations with exogenous, where ApolloPFN is SOTA or competitive. The authors note the Moirai confound: training exposure to public benchmarks complicates strict zero-shot comparisons—where the transparency is appreciated.


* A comprehensive table on classical univariate shows ApolloPFN is competitive too, though the conceptual focus is on the setting with an increased number of exogenous variables

**Weaknesses:**

I believe the paper has a limited analysis of probabilistic calibration and robustness to exogenous shift, which can be needed in real-life time-series settings. I also believe that the heavy reliance on synthetic priors deserves more discussion on alignment to real exogenous processes. I agree with the scaling constraints from quadratic attention as noted by the authors, which is a considerable weakness.

**Questions:**

- Can the authors provide calibration (PIT, coverage) and counterfactual sensitivity analyses for exogenous covariates (e.g., price elasticity sanity checks)?


- Is it possible to add ablations isolating each time-aware architectural change and the synthetic prior components that matter most?


- Is it feasible to design and evaluate robustness under exogenous distribution shift (e.g., unseen promo patterns)?


- Can you clarify zero-shot rigor, e.g., ensuring no leakage from synthetic priors into test covariate regimes; specifying any exogenous data preprocessing rules?

---

> ### Author Response · Authors · 2025-11-22
>
> We appreciate your enthusiasm for our work acknowledging that ApolloPFN is a meaningful advance. We now address your questions.
>
> __On probabilistic calibration.__ We believe that our probabilistic predictions are reasonably calibrated based on our strong sCRPS results which is a metric sensitive to calibration errors (Appendix D). However, indeed it is no substitute for a direct metric like a PIT test. To this end, we have now included in Appendix E that analysis in Figure 7. For the first quantiles the distribution is not completely uniform  but it is uniform for the rest of the quantiles.
>
> __On exogenous shift and zero-shot rigor.__ Your question about the robustness to exogenous shifts lies at the crux of why our synthetic data generation procedure performs so well in practice. The sampled SCMs are actually unrelated to the task at hand. For example, we didn’t explicitly add promotional spikes or price phase shifts; yet, the model is able to pick that behavior in-context when making predictions. In other words, our synthetic data generation does not encode any of the behaviors that we expect to see in practice but it performs well suggesting that we are robust to exogenous shifts. Our experience when training ApolloPFN is that once the model has seen a sufficient number of hard examples during training it learns a favorable mechanism to make predictions that actually work well in practice. Yet, this does not mean that ApolloPFN will not break at some sufficiently large exogenous shift.
>
> __On quadratic attention.__ In terms of reducing the quadratic costs in attention, we do find it an important avenue of future research for dealing with long contexts or more features. Currently our modifications have focused on making the model improve performance but reducing cost is also relevant in practice. However, we would like to emphasize that currently finding relevant exogenous variables is a challenge and our model can already handle up to 128.
>
> __On ablations for architectural modifications.__ In Figure 4 we probed the effects of adding each of the different modifications one step at a time. Indeed, as seen in the figure, across several datasets we see a consistent monotonic increase in performance when adding the different components. However, based on your suggestion we have run additional ablations also on the type of positional encoding to understand its effect as seen in Table 7 in the new Appendix E.
>
> Please let us know if we have addressed your questions properly. Additionally, as we perceive that you have a solid understanding of our work, we would like to ask you if during the discussion period you could address any misunderstandings or misconceptions that other reviews might have.

---

### Meta-Review · Area_Chair_kg9r · 2026-01-06

**Summary:**

This paper empowers the capability of time series foundation models to the prevalent zero-shot forecasting scenario with exogenous variables. The main finding is that TabPFN, a tabular foundation model published in Nature, lacks temporal inductive bias modeling capability. The proposed solution, ApolloPFN, constitutes two relatively minor modifications to TabPFN: (1) a temporal synthetic data generation procedure and (2) architectural modifications adding RoPE positional encodings and expanded attention. Results show sharp improvements, but the used datasets and baselines are relatively incomprehensive.

Reviewers generally agree that the studied scenario is worthwhile for real-world problems, but the key finding appears to be a hindsight that TabPFN needs temporal adaptations, and the technical innovations are relatively minor. Factors like method scalability and writing quality also drag the overall score to fall below the acceptance threshold.

**Reviewer Concerns:**

Reviewer c4ks: Comments addressed well.
Reviewer iujA: Comments addressed well.
Reviewer pg4u: Concerns on the limits for T (#time steps) and F (#features) and the ability to handle variable-length sequences. Not addressed in a convincing way.
Reviewer hrkC: Why should random DAGs with MLPs/decision trees as node functions capture real forecasting scenarios? Not addressed well.

**Reviewer Scores:**

Reviewers generally won't change their original scores by witnessing the rebuttal.

---

### Decision · Program_Chairs · 2026-01-26

Reject